# RNA-Seq Provides Insights into VEGF-Induced Signaling in Human Retinal Microvascular Endothelial Cells: Implications in Retinopathy of Prematurity

**DOI:** 10.3390/ijms23137354

**Published:** 2022-07-01

**Authors:** Aniket Ramshekar, Colin A. Bretz, M. Elizabeth Hartnett

**Affiliations:** Department of Ophthalmology and Visual Sciences, John A. Moran Eye Center, University of Utah, 65 Mario Capecchi Dr, Salt Lake City, UT 84132, USA; u1088294@utah.edu (A.R.); u6003023@utah.edu (C.A.B.)

**Keywords:** ROP, VEGF, VEGFR2, KDR, STAT3, RNA-seq, HRMECs

## Abstract

The pathophysiology of retinopathy of prematurity (ROP) is postulated to first involve delayed intraretinal vascularization, followed by intravitreal neovascularization (IVNV). Although intravitreal agents that reduce the bioactivity of vascular endothelial growth factor (VEGF) are used to treat IVNV, concerns exist regarding their effects on intraretinal vascularization. In an experimental ROP model, VEGF receptor 2 (VEGFR2) knockdown in retinal endothelial cells reduced IVNV and promoted intraretinal vascularization, whereas knockdown of a downstream effector, signal transducer and activator of transcription 3 (STAT3) in retinal endothelial cells only reduced IVNV. In this study, we tested the hypothesis that the different pathways involved in VEGF-triggered VEGFR2 signaling and VEGF-triggered STAT3 signaling in retinal endothelial cells would allow us to delineate signaling pathways involved in IVNV from those involved in intraretinal vascularization in ROP. To address our hypothesis, we used RNA-sequencing and pathway enrichment analysis to determine changes in the transcriptome of cultured human retinal microvascular endothelial cells (HRMECs). Of the enriched pathways, inactivation of oncostatin M signaling was predicted by either *KDR* or *STAT3* knockdown in the presence of VEGF. Activation of kinetochore metaphase signaling was predicted by *KDR* knockdown, whereas inactivation was predicted by *STAT3* knockdown in the presence of VEGF. Inactivation of signaling by the Rho family of GTPases was predicted by *KDR* knockdown, but activation was predicted by *STAT3* knockdown in the presence of VEGF. Taken together, our data identified unique signaling pathway differences between VEGF-triggered VEGFR2 and VEGF-triggered STAT3 in HRMECs that might have implications in ROP.

## 1. Introduction

Retinopathy of prematurity (ROP) remains a leading cause of vision loss and blindness in children worldwide [1]. The pathophysiology is described by a two-phase hypothesis in which Phase I is characterized by compromised and delayed intraretinal vascularization, and Phase II is characterized by vision-threatening extraretinal or intravitreal neovascularization (IVNV) [2]. Current approaches to treat Phase II ROP include laser treatment to ablate regions of the peripheral avascular retina [3] or intravitreal administration of agents that reduce the bioactivity of vascular endothelial growth factor (VEGF) [4]. However, intravitreal anti-VEGF agents at certain doses have been associated with later reactivation of treatment-warranted ROP from the persistent avascular retina in human studies [5,6,7]. Furthermore, non-specific inhibition of VEGF has been associated with increased retinal cell death in mice [8] and rats [9,10], emphasizing the importance of VEGF in the survival of endothelial, neural, and glial cells. Therefore, it is important to understand approaches to regulate VEGF-mediated cellular and molecular events in specific cell types in order to inhibit IVNV and extend intraretinal vascularization to the ora serrata.

VEGF interacts with several receptors and co-receptors on different cell types [11]. Specifically, VEGF-triggered signaling through VEGF receptor 2 (VEGFR2) in endothelial cells has been demonstrated to mediate both pathologic and physiologic angiogenesis [12]. We used the most representative animal model of ROP, the rat oxygen-induced retinopathy (OIR) model [13], to understand the role of endothelial VEGFR2 in ROP pathology. In the rat OIR model, subretinal delivery of lentiviral vectors that expressed shRNA targeting *Kdr* driven by a *Cdh5* promoter reduced IVNV and promoted intraretinal vascularization compared to subretinal delivery of lentiviral vectors that expressed shRNA targeting luciferase driven by a *Cdh5* promoter as control [14]. This finding suggested that regulation of VEGFR2 activation in retinal endothelial cells could inhibit pathologic angiogenesis and promote physiologic angiogenesis. Therefore, we assessed downstream effectors of VEGF-triggered signaling to target IVNV and extend intraretinal vascularization. In previous studies using the rat OIR model, we found evidence that the oxygen fluctuation-induced stress in the rat OIR model increased VEGF-triggered STAT3 activation by signaling through the JAK/STAT pathway, and that an intravitreal JAK2 inhibitor reduced IVNV and promoted intraretinal vascular extension [15,16]. We therefore postulated that retinal endothelial STAT3 was a downstream effector of VEGF-triggered signaling that promoted the development of IVNV and delayed the extension of intraretinal vascularization. In the rat OIR model, however, subretinal delivery of lentiviral vectors that expressed shRNA targeting *Stat3* driven by a *Cdh5* promoter reduced IVNV but did not extend intraretinal vascularization compared to subretinal delivery of lentiviral vectors that expressed shRNA targeting luciferase driven by a *Cdh5* promoter as control [14]. These findings suggested that VEGF-triggered STAT3 activation in retinal endothelial cells is necessary for IVNV, but not for intraretinal vascularization.

In this study, we employed high-throughput RNA sequencing (RNA-seq) to address a hypothesis that the different pathways involved in VEGF-triggered VEGFR2 signaling and VEGF-triggered STAT3 signaling in human retinal endothelial cells (HRMECs) would allow us to delineate signaling pathways involved in IVNV from those involved in intraretinal vascularization.

## 2. Results

### 2.1. Quality Control of Biologic Samples in RNA-Seq Analysis

In order to assess sufficient knockdown of either STAT3 or VEGFR2, we measured *STAT3* or *KDR* mRNA expression by RT-PCR in HRMECs 24 or 36 h after siRNA transfection. At 24 h and 36 h after transfection and compared to respective timepoint controls, HRMECs transfected with STAT3 siRNA had significantly reduced *STAT3* mRNA expression (*p*-value < 0.05, Figure 1A), while HRMECs transfected with KDR siRNA had significantly reduced *KDR* mRNA expression (*p*-value < 0.05, Figure 1B). These findings provided us with a suitable time frame to treat transfected HRMECs.

We then treated control siRNA transfected HRMECs with VEGF for four hours and assessed several VEGF-mediated gene targets (e.g., *VCAM1*, *BCL2*, *IL6*, and *CCND1*), all of which had significantly increased mRNA expression compared to volume-matched vehicle control (*p*-value < 0.05, Figure 1C). These findings suggested that four hours of VEGF treatment was sufficient to induce gene expression changes. We next evaluated the RNA integrity value (RIN) to ensure that the RNA isolated from HRMECs was suitable for high-throughput RNA-seq. The RIN for all samples used in the STAT3 and KDR RNA-seq experiments was greater than 8.0 and confirmed that the isolated RNA samples were of high quality (Appendix A) [17].

After high-throughput sequencing, a principal component analysis (PCA) was performed using the regularized log-transformed (Rlog) count values for the top 500 most variable genes. In the STAT3 RNA-seq dataset, PCA identified two principal components (PCs) that accounted for 87.5% of the variance (Figure 1D). PC1 separated biologic replicates treated with PBS from biologic replicates treated with VEGF (78.4% variance), and PC2 separated biologic replicates transfected with STAT3 siRNA from biologic replicates transfected with control siRNA (9.1% variance). Biologic replicates that were under the same treatment conditions were visually grouped together and provided support for the use of all samples in the STAT3 RNA-seq dataset. In the KDR RNA-seq dataset, PCA identified two PCs that accounted for 87.8% of the variance (Figure 1E): PC1 separated control siRNA-transfected biologic replicates treated with PBS from control siRNA-transfected biologic replicates treated with VEGF and KDR siRNA-transfected biologic replicates from control siRNA-transfected biologic replicates (64.9% variance). Notably, PCA identified grouping of KDR siRNA-transfected biologic replicates regardless of VEGF or PBS treatment. Nonetheless, the biologic replicates that were under the same treatment conditions were visually grouped together and provided support for the use of all samples in the KDR RNA-seq dataset.

### 2.2. VEGF-Mediated Gene Regulation in HRMECs

VEGF-triggered signaling is implicated in the physiologic and pathologic development of retinal blood vessels in ROP [18]. We therefore compared the transcriptomes between VEGF-treated and PBS-treated HRMECs transfected with control siRNA to determine the differentially expressed genes. Since the two groups were sequenced in the STAT3 RNA-seq and the KDR RNA-seq datasets, we analyzed and compared VEGF-mediated differential gene expression in both datasets using DESeq2. Compared to PBS, VEGF differentially regulated 2802 genes (1590 upregulated genes and 1212 downregulated genes, adjusted *p*-value < 0.05) in HRMECs transfected with control siRNA in the STAT3 RNA-seq dataset (Appendix A), whereas VEGF differentially regulated 2171 genes (1362 upregulated genes and 809 downregulated genes, adjusted *p*-value < 0.05) in HRMECs transfected with control siRNA in the KDR RNA-seq dataset (Appendix A). In both datasets, VEGF treatment was associated with a greater fold increase than a decrease of the differentially expressed genes (Figure 2A,B).

To identify pathway enrichments associated with the VEGF-mediated genes, we used the Canonical Pathways feature in IPA. IPA identified the enrichment of 222 signaling pathways in the STAT3 RNA-seq dataset (Appendix A, adjusted *p*-value < 0.05) and 233 signaling pathways in the KDR RNA-seq dataset (Appendix A, adjusted *p*-value < 0.05). Furthermore, IPA identified enriched signaling pathways that were predicted to be significantly activated (*n* = 10, z-score ≥ 2) or inactivated (*n* = 2, z-score ≤ −2) in the STAT3 RNA-seq dataset (Figure 2C), and significantly activated (*n* = 46, z-score ≥ 2) or inactivated (*n* = 5, z-score ≤ −2) in the KDR RNA-seq dataset (Figure 2D). Of the enriched signaling pathways, unfolded protein response, regulation of epithelial mesenchymal transition by growth factors pathway, insulin secretion signaling pathway, tumor microenvironment pathway, UDP-N-acetyl-D-glucosamine biosynthesis II, ERK5 signaling, oncostatin M signaling, and FAT10 cancer signaling pathway were predicted to be significantly activated in both datasets. However, we also observed that some of the VEGF-mediated signaling pathways were only significantly enriched in one of the datasets. Furthermore, we observed that only 31% (1184) of the differentially expressed genes were shared between the two comparative analyses (Figure 2E). We speculated that the differences in VEGF-mediated gene regulation between the two datasets might be due to differences in the RNA-seq methods. To address this consideration, we used the Analysis Match feature in IPA and found that the comparative analyses (VEGF versus PBS in control siRNA transfected HRMECs) in the STAT3 and KDR RNA-seq datasets were very similar and in fact the most similar out of ~90,000 comparative transcriptome studies in IPA (overall z-score = 58.82). Taken together, the data suggested that VEGF-mediated gene regulation in our two datasets was sufficiently similar to compare despite the differences in RNA-seq methods.

### 2.3. VEGFR2-Mediated Gene Regulation in HRMECs

In the rat OIR model, knockdown of VEGFR2 in retinal endothelial cells by shRNA targeting *Kdr* was associated with decreased IVNV and extension of intraretinal vascularization compared to littermate controls treated with luciferase shRNA [14]. We therefore compared the sequenced transcriptome between VEGF-treated HRMECs transfected with KDR siRNA to VEGF-treated HRMECs transfected with control siRNA in the KDR RNA-seq dataset. In the presence of VEGF treatment, knockdown of *KDR* differentially expressed 3765 genes (1460 upregulated and 2305 downregulated, adjusted *p*-value < 0.05) compared to control (Figure 3A, Appendix A). IPA identified the significant enrichment of 309 canonical pathways (adjusted *p*-value < 0.05, Appendix A) associated with the differentially expressed genes. IPA predicted 134 of the enriched canonical pathways to be significantly inactivated (z-score ≤ −2) and 14 to be significantly activated (z-score ≥ 2, Appendix A). Notably, canonical pathways that were significantly activated by VEGF (e.g., insulin secretion signaling pathway, tumor microenvironment pathway, oncostatin M signaling, ERK5 signaling, regulation of epithelial-mesenchymal transition by growth factors pathway, and unfolded protein response) were predicted to be significantly inactivated (z-score ≤ −2) with *KDR* knockdown. Also, pathways implicated in cell movement (e.g., erythropoietin signaling pathway, ERB2-ERBB3 signaling, PDGF signaling, JAK/STAT signaling, EGF signaling, signaling by Rho family GTPases, and HIF1α signaling) were also predicted to be significantly inactivated (z-score ≤ −2). IPA, however, predicted that pathways involved in cell cycle regulation (e.g., kinetochore metaphase signaling and chromosomal replication) were significantly activated (z-score ≥ 2). Taken together, the pathway enrichment data suggested that VEGFR2 is necessary to mediate genes implicated in several signaling pathways in addition to VEGF-triggered signaling.

In support of the above interpretation, we only observed a 30% (1360 genes) overlap between the VEGF-induced VEGFR2-mediated genes and VEGF-mediated genes (Figure 3B). We, therefore, sought to understand the relative gene expression patterns of the 3765 genes differentially regulated by VEGF-induced VEGFR2 across all four of the experimental groups in the KDR RNA-seq dataset using a data-driven approach. Hierarchical clustering identified four visual gene expression patterns (Figure 3C, row dendrogram). Regardless of VEGF or PBS treatment, knockdown of *KDR* upregulated cluster 2 genes (*n* = 1210) or downregulated cluster 3 genes (*n* = 836). Knockdown of *KDR* prevented VEGF-induced downregulation of cluster 1 genes (*n* = 250) or upregulation of cluster 4 genes (*n* = 1469). Notably, 208 of the cluster 1 genes (83%), 151 of the cluster 2 genes (13%), 22 of the cluster 3 genes (3%), and 979 (67%) of the cluster 4 genes were also differentially regulated by VEGF (Figure 3D). Taken together, the data suggested that knockdown of *KDR* differentially regulated genes that did not overlap with those involved in VEGF-triggered signaling. We next determined pathway enrichments associated with the genes in each cluster using IPA. IPA identified the significant enrichment of two signaling pathways associated with cluster 1 genes, 45 signaling pathways associated with cluster 2 genes, 48 signaling pathways associated with cluster 3 genes, and 269 signaling pathways associated with cluster 4 genes (Appendix A). IPA did not make any significant pathway activation status predictions of the enriched signaling pathways associated with cluster 1 genes. Of the significantly enriched pathways associated with cluster 2 genes, IPA predicted significant inactivation of the sumoylation pathway (z-score = −2.324) and significant activation (z-score ≥ 2) of pathways related to cell cycle regulation (e.g., cell cycle control of chromosomal replication, kinetochore metaphase signaling, mitotic roles of polo-like kinase, and estrogen-mediated S phase entry), fatty acid and lipid metabolism (e.g., cholesterol biosynthesis, 3-phosphoinositide biosynthesis, 3-phosphoinositide degradation), and sugar derivatives metabolism (e.g., D-myo-inositol (1,4,5,6)-tetrakisphosphate biosynthesis and D-myo-inositol (3,4,5,6)-tetrakisphosphate biosynthesis, D-myo-inositol-5-phosphate metabolism) in response to VEGF treatment in HRMECs with *KDR* knocked down compared to control (Figure 3E). Of the significantly enriched pathways associated with cluster 3 genes, IPA predicted significant activation of the PPARα/RXRα transcriptional complex and significant inhibition of pathways related to cellular stress and injury (e.g., CLEAR signaling pathway, autophagy, pulmonary fibrosis idiopathy signaling pathway, and GP6 signaling pathway), cellular morphology and proliferation (e.g., EGF signaling, erythropoietin signaling, PDGF signaling, RAC signaling, senescence pathway, and the hepatic fibrosis signaling pathway), and cytokine signaling (e.g., FLT3 signaling, IL-15 production, and IL-1 signaling) in response to VEGF treatment in HRMECs with *KDR* knocked down compared to control (Figure 3F). Of the significantly enriched pathways associated with cluster 4 genes, IPA predicted significant activation or inhibition of 191 different pathways (Appendix A). Notably, pathways related to VEGF signaling were predicted to be significantly inactivated in response to VEGF in HRMECs with *KDR* knocked down compared to control. Taken together, the data provided insight into pathways regulated by VEGFR2 in the presence and absence of VEGF stimulation.

### 2.4. STAT3-Mediated Gene Regulation in HRMECs

In the rat OIR model, we previously observed that knockdown of STAT3 in retinal endothelial cells by shRNA targeting *Stat3* was associated with significantly reduced IVNV compared to littermate controls treated with luciferase shRNA [14]. We therefore compared the sequenced transcriptomes between VEGF treated HRMECs transfected with STAT3 siRNA and VEGF treated HRMECs transfected with control siRNA in the STAT3 RNA-seq dataset. In the presence of VEGF, knockdown of *STAT3* was associated with the differential expression of 348 genes (189 upregulated genes and 159 downregulated genes, adjusted *p*-value < 0.05) compared to control (Figure 4A, Appendix A). IPA identified significant enrichment of 143 canonical pathways (adjusted *p*-value < 0.05, Appendix A) associated with the differentially expressed genes by VEGF-induced STAT3. Notably, kinetochore metaphase signaling and oncostatin M signaling were predicted to be significantly inactivated (z-score ≤ −2), while ATM signaling and signaling by Rho Family GTPases were predicted to be significantly activated (z-score ≥ 2, Figure 4B).

We observed that only 156 out of 348 genes of the VEGF-induced STAT3-mediated genes were also differentially expressed by VEGF in the same dataset (Figure 4C). We therefore visualized the relative gene expression patterns of the 348 differentially expressed genes by VEGF-induced STAT3 across all four experimental groups using a data-driven approach. Hierarchical clustering of the differentially expressed genes resulted in the visual identification of four different gene expression patterns (Figure 4D, row dendrogram). Regardless of VEGF or PBS treatment, knockdown of *STAT3* downregulated cluster 2 genes (*n* = 116) and upregulated cluster 3 genes (*n* = 71) compared to control. Compared to control, knockdown of *STAT3* reduced the VEGF-induced upregulation of cluster 1 genes (*n* = 43) and reduced VEGF-induced downregulation of cluster 4 genes (*n* = 118). We then assessed pathway enrichment associated with the genes from each cluster. IPA did not identify significant pathway enrichment associated with cluster 1 genes (adjusted *p*-value > 0.05) but did identify significant pathway enrichment (adjusted *p*-value < 0.05) of 42 pathways associated with cluster 2 genes, 24 different pathways associated with cluster 3 genes, and four different pathways associated with cluster 4 genes (Appendix A). Of the significantly enriched pathways associated with cluster 2 genes, pathways involved in DNA damage checkpoint (e.g., G2/M DNA damage checkpoint regulation and ATM signaling) were predicted to be significantly activated (z-score ≥ 2), whereas pathways involved in cell cycle regulation (e.g., kinetochore metaphase signaling and cyclin-mediated signaling) and cellular movement (e.g., IL-3 signaling, PDGF signaling, and ephrin receptor signaling) were predicted to be significantly inactivated (z-score ≤ −2) in VEGF-treated HRMECs with *STAT3* knocked down compared to control (Figure 4E). Of the significantly enriched pathways associated with cluster 3 genes, pathways involved in cellular movement (e.g., pulmonary healing signaling pathway and IL-8 signaling) and cellular stress (pulmonary fibrosis idiopathic signaling pathway and hepatic fibrosis signaling pathway) were predicted to be significantly activated (z-score ≥ 2) in VEGF- treated HRMECs with *STAT3* knocked down compared to control (Figure 4F). There were no significant activation status predictions made by IPA for the enriched pathways associated with cluster 4 genes (z-score ≤ |2|). Taken together, the data suggested that STAT3-mediated differential gene regulation occurs in the presence or absence of VEGF-triggered signaling.

### 2.5. Identification of Canonical Pathways Implicated in STAT3-Mediated or VEGFR2-Mediated Signaling by VEGF

To delineate pathways implicated in IVNV or intraretinal vascular extension, we compared the enriched canonical pathways associated with VEGF-induced STAT3-mediated genes to those associated with VEGF-induced VEGFR2-mediated genes in HRMECs. We observed 180 unique pathways associated with VEGF-induced VEGFR2-mediated genes, 14 unique pathways associated with VEGF-induced STAT3-mediated genes, and 129 pathways that were common in both comparative analyses. We further analyzed the common enriched pathways to determine differences in pathway activation predictions by IPA. IPA predicted significant inactivation of oncostatin M signaling in response to VEGF in HRMECs with *STAT3* (z-score = −2) or *KDR* knocked down (z-score = −3.3) compared to the respective dataset controls. Compared to respective dataset controls, IPA predicted the significant inactivation of kinetochore metaphase signaling in response to VEGF in HRMECs with *STAT3* knocked down (z-score = −2.828), but significant activation of this signaling pathway in response to VEGF in HRMECs with *KDR* knocked down (z-score = 2.121). Compared to the respective dataset controls, IPA predicted significant activation of signaling by Rho family GTPases in response to VEGF in HRMECs with *STAT3* knocked down (z-score = 2.121), but significant inactivation of this signaling pathway in response to VEGF in HRMECs with *KDR* knocked down (z-score = −3.046). Taken together, the data suggested differences in pathway activation between the VEGF-induced STAT3-mediated genes and VEGF-induced VEGFR2-mediated genes.

## 3. Discussion

In children, ROP remains a leading cause of vision loss and blindness worldwide [1]. The pathophysiology involves delayed peripheral intraretinal vascularization and compromised physiologic vascularity from oxygen stresses initially followed by the development of extraretinal or IVNV [2]. One approach to treat IVNV is the administration of intravitreal anti-VEGF agents [19,20,21,22,23]. Although experimental studies have provided evidence that optimal anti-VEGF effects using gene therapy to knockdown the expression of VEGF can reduce IVNV without interfering with intraretinal vascularization, concerns exist regarding the effects of VEGF inhibition on the neural retina as well as its potential effects on developing organs once leaked into the systemic circulation [9,24]. Understanding VEGF-triggered signaling in cell types implicated in the development of IVNV is necessary to develop more targeted therapeutic approaches. Previously, we found that regulating VEGFR2 signaling in retinal endothelial cells by gene-therapy approaches was associated with reduced IVNV and increased peripheral intraretinal vascularization in the rat OIR model, a translational model that recapitulates aspects of human ROP [13]; however, we found that knockdown of STAT3, a downstream effector of VEGF-triggered signaling, in retinal endothelial cells was associated with reduced IVNV in the rat OIR model but not increased intraretinal peripheral vascularization [14]. Therefore, this study sought to gain insight into the different signaling pathways regulated by VEGF-triggered VEGFR2 compared to VEGF-triggered STAT3 in cultured HRMECs.

We found that VEGF differentially regulated 2802 genes in the STAT3 RNA-seq dataset or 2191 genes in the KDR RNA-seq dataset; several of these genes are involved in angiogenic pathways. In the control siRNA transfected HRMECs, we observed significant enrichment of pathways involved in senescence, HIF1α signaling, PI3K/AKT signaling, JAK/STAT signaling, ERK/MAPK signaling, IGF-1 signaling, oncostatin M signaling, and VEGF signaling in response to VEGF compared to vehicle control in both RNA-seq datasets. A subset of these enriched pathways was also predicted to be significantly activated by IPA in both datasets (e.g., oncostatin M signaling and ERK5 signaling). Notably, some of these pathways were also associated with whole retinal transcriptomes of retinas from mice raised in different OIR models compared to room air-raised controls. For example, enrichment of the senescence pathway [25], P13K/AKT signaling pathway [26], hypoxia signaling [27], and VEGF receptor signaling [28] was identified in OIR-raised mouse retinas compared to age-matched, room air-raised mouse retinas at postnatal day (p)17—a time point in the murine OIR model associated with the development of maximal IVNV [29]. Taken together, the data suggest that the VEGF-mediated transcriptome changes in cultured HRMECs have similarities to OIR-mediated transcriptome changes in retinas isolated from mice and highlight the potential translational implications of this RNA-seq dataset to human ROP.

We previously observed that VEGFR2-triggered signaling promoted the development of IVNV and delayed peripheral intraretinal vascularization in the rat OIR model [14]. We, therefore, compared VEGF-treated HRMECs transfected with KDR siRNA to VEGF- treated HRMECs transfected with control siRNA. In the presence of VEGF treatment, we found that *KDR* knockdown differentially expressed 3765 genes; however, only 1360 of these genes were also differentially expressed by VEGF in the same dataset. These findings suggested that VEGFR2-mediated transcriptome changes also occur independently of VEGF-triggered signaling. We then evaluated pathway enrichment using IPA and observed significant enrichment and predicted activation by IPA of cell cycle regulation pathways (e.g., kinetochore metaphase signaling) associated with the VEGFR2-mediated genes. These findings might provide a plausible mechanistic insight into increased intraretinal vascularization in OIR rat pups with VEGFR2 knocked down compared to control. We also observed significant enrichment and predicted inactivation by IPA of pathways involved in cell migration (e.g., signaling by Rho family of GTPases, PDGF signaling, PI3K signaling, interleukin signaling, and erythropoietin signaling) and VEGF-triggered signaling (e.g., PI3K/AKT signaling, ERK signaling, HIF1α signaling, oncostatin M signaling, etc.). The predicted inactivation of these pathways might be implicated in the reduced development of VEGFR2-mediated IVNV in the rat OIR model. In support of this notion, intravitreal inhibition of ERK signaling reduced IVNV without affecting intraretinal vascular development at p20 compared to intravitreal vehicle control in the rat OIR model [30]. Further studies in the rat OIR model will help to define the role of the other enriched pathways in either physiologic or pathologic angiogenesis implicated in ROP.

We previously observed that STAT3 was necessary for the development of IVNV in the rat OIR model but not for intraretinal vascular extension [14]. In this study, we found that *STAT3* knockdown led to the differential expression of 348 genes in the presence of VEGF; however, only 156 of these genes were also differentially expressed by VEGF in the same dataset. These findings support our hypothesis that VEGF-triggered STAT3-mediated transcriptome changes occur independently of VEGF-triggered VEGFR2 signaling. To delineate signaling pathways involved with IVNV from those involved in intraretinal vascular extension, we compared the enriched pathways associated with VEGF-induced STAT3-mediated genes to those associated with VEGF-induced VEGFR2-mediated genes. We observed 129 pathways that were common to both VEGF-induced STAT3-mediated genes and VEGF-induced VEGFR2-mediated genes. Notably, we observed significant enrichment and predicted inactivation by IPA of oncostatin M signaling associated with VEGF-induced STAT3-mediated genes and VEGF-induced VEGFR2-mediated genes in HRMECs. This finding might implicate oncostatin M signaling in the development of IVNV. However, there was limited literature to support the role of oncostatin M signaling in OIR models. We also observed significant enrichment and predicted the inactivation of kinetochore metaphase signaling associated with the VEGF-induced STAT3-mediated genes, but significant enrichment and predicted activation of this pathway associated with the VEGF-induced VEGFR2-mediated genes. This finding might implicate kinetochore metaphase signaling in regulating intraretinal vascular extension. In support of this notion, kinetochore protein, Spc25, was significantly reduced in OIR mice compared to room air controls at p17 [31]. Further studies regarding the role and pharmacologic regulation of the kinetochore metaphase signaling pathway and oncostatin M signaling in the rat OIR are warranted. Finally, we observed significant enrichment and predicted activation of signaling by Rho family GTPases associated with VEGF-induced STAT3-mediated genes, but predicted the inactivation of this signaling pathway associated with VEGF-induced VEGFR2-mediated genes. This finding might implicate signaling by Rho family GTPases in regulating peripheral intraretinal vascularization. For example, disrupting the activation of Cdc42 or RhoJ GTPases in mice resulted in dysregulated intraretinal vascular extension to the ora serrata [32]. However, administration of an intravitreal Rho-kinase inhibitor has been demonstrated to significantly reduce IVNV compared to intravitreal vehicle control at p17 in the mouse OIR model [33]. These findings suggest that Rho GTPases are involved in physiologic and pathologic angiogenesis.

In conclusion, our data provided insight into plausible differential VEGF-mediated signaling pathways between VEGFR2 and STAT3. Additional studies are required in cell and animal models, including in representative models like the rat OIR, to validate and/or test various pathways in IVNV and peripheral intraretinal vascularization related to ROP.

## 4. Materials and Methods

### 4.1. Cell Culture Conditions

HRMECs (Cell Systems, Krikland, WA, USA), passages 3–5, were cultured in attachment factor (Cell Systems) coated cultureware with the Endothelial Growth Medium BulletKit (EGM, Lonza, Walkersville, MD, USA) supplemented with 10% fetal bovine serum (FBS). We routinely verified in parallel experiments that HRMECs maintained endothelial phenotypes (Appendix A).

### 4.2. Transfection

HRMECs were transfected with equal concentrations of control small interfering RNA (siRNA), STAT3 siRNA, or KDR siRNA oligos (Thermo Fisher Scientific, Waltham, MA, USA) using Targefect Solution A and Virofect (Targeting Systems, El Cajon, CA, USA) as previously described [34].

### 4.3. Treatments and RNA Isolation

Twenty-four hours after transfection, HRMECs were serum starved in Endothelial Basal Media (EBM, Lonza) for eight hours and then treated with either VEGF (25 ng/mL, R&D Systems, Minneapolis, MN, USA) or an equal volume of phosphate-buffered saline (PBS) as vehicle control for four hours. After treatment, cells were lysed in buffer RLT supplemented with β-mercaptoethanol (1:100, Sigma-Aldrich, St. Louis, MO, USA) and RNA was isolated from lysates using the Qiagen RNeasy Kit following the protocol provided by the manufacturer (Qiagen, Valencia, CA, USA).

### 4.4. RNA Quality Assessment

RNA quality assessment was performed by the High-Throughput Genomics Core Facility (Huntsman Cancer Institute, University of Utah, Salt Lake City, UT, USA). RNA concentration was measured using a Qubit RNA BR Assay Kit (Thermo Fisher Scientific, Waltham, MA, USA). RNA quality was evaluated using an RNA ScreenTape Assay (Agilent Technologies, Santa Clara, CA, USA).

### 4.5. Library Preparation and High-Throughput RNA Sequencing

Library preparation and high-throughput RNA sequencing (RNA-seq) was performed by the High-Throughput Genomics Core Facility (Huntsman Cancer Institute, University of Utah, Salt Lake City, UT, USA).

For the STAT3 RNA-seq dataset, intact poly(A) RNA was purified from total RNA samples and libraries were prepared using the TruSeq Stranded mRNA Library Preparation Kit (Illumina, San Diego, CA, USA). Purified libraries were qualified using a DNA ScreenTape Assay (Agilent Technologies). The molarity of adapter-modified molecules was quantified by quantitative PCR using the Kapa Library Quantification Kit (Roche Sequencing and Life Science, Indianapolis, IN, USA). Individual libraries were normalized to 5 nM and equal volumes were pooled in preparation for sequence analysis. Sequencing libraries were chemically denatured and applied to an Illumina HiSeq v4 single read flow cell using an Illumina cBot. Hybridized molecules were clonally amplified and annealed to sequencing primers with reagents from an Illumina HiSeq SR Cluster Kit v4-cBot (Illumina). Following transfer of the flow cell to the Illumina HiSeq 2500 instrument, a 50-cycle single-read sequence run was performed using HiSeq SBS Kit V4 Cycle Kit (Illumina).

For the KDR RNA-seq dataset, intact poly(A) RNA was purified from total RNA samples using the NEBNext Poly(A) mRNA Magnetic Isolation Module (New England BioLabs, Ipswich, MA, USA), and libraries were prepared using the NEBNext Ultra II Directional RNA Library Prep Kit (New England BioLabs). Purified libraries were qualified using a DNA ScreenTape Assay (Agilent Technologies). The molarity of adapter-modified molecules was quantified by quantitative PCR using the Kapa Library Quant Kit (Roche Sequencing and Life Science). Individual libraries were normalized to 5 nM in preparation for sequence analysis. Sequencing libraries were chemically denatured and applied to an Illumina NovaSeq flow cell using the NovaSeq XP workflow (Illumina). Following transfer of the flow cell to the Illumina NovaSeq 6000 instrument, a 150 × 150 cycle paired end sequence run was performed using a NovaSeq 6000 S4 Reagent Kit v1.5 (Illumina).

### 4.6. Genome Alignment and Differential Gene Expression Analysis

Genome alignment and differential expression was performed by the Bioinformatic Analysis Shared Resource (Huntsman Cancer Institute, University of Utah, Salt Lake City, UT, USA).

For the STAT3 RNA-seq dataset, sequences were aligned to the hg38 genome and splice junctions to known transcripts (RefSeq, radius 46 bp) using novoalign (v2.8.1, NovoCraft, Selangor, Malaysia), allowing for up to 50 alignments per read and trimming sequencing adapters. Alignments to splice junctions were converted back to genomic coordinates adapters with USeq SamTranscriptomeParser (v8.8.8), allowing for one alignment per read. Alignment data were summarized in read count matrices for each submitted biologic sample.

For the KDR RNA-seq dataset, optical duplicates were removed with Clumpify (v38.34), adapters were trimmed using Cutadapt (v2.8), genome alignment was performed using STAR (v2.7.9a) to the hg38 genome, and genes were counted using featureCounts (v1.6.3) Alignment data were summarized in read count matrices for each submitted biologic sample.

Differentially expressed genes between two experimental groups in the same RNA-seq dataset were determined using the R software packages, hciR and DESeq2 (v1.32.0) [35], with the read counts matrices as the input data. An adjusted *p*-value < 0.05 calculated with the Wald test with a Benjamini-Hochberg multiple testing correction in DESeq2 was used as the significance threshold to determine differentially expressed genes.

Hierarchical clustering was performed with Ward’s linkage method and the Euclidean distance metric on scaled regularized log (RLog) transformed read counts using pheatmap (v1.0.12).

### 4.7. Pathway Enrichment Analysis

A list of differentially expressed genes from each comparison was uploaded to Ingenuity Pathway Analysis software (IPA, Qiagen, Redwood City, CA, USA). The Canonical Pathway tool was used to determine pathway enrichment and to predict pathway activation states, based on published literature, associated with the list of differentially expressed genes between the compared experimental groups. An adjusted *p*-value < 0.05 calculated with a right-tailed Fisher’s exact test followed by the Benjamini-Hochberg multiple testing correction was used as the significance threshold for pathway enrichment. A z-score ≥ |2| was used as the significance threshold for the pathway activation status prediction [36].

### 4.8. Real-Time PCR Analysis

Isolated RNA samples from transfected and treated HRMECs were reverse transcribed to obtain cDNA using the High Capacity cDNA Reverse Transcription Kit following the protocol provided by the manufacturer (Thermo Fisher Scientific, Waltham, MA, USA). The cDNA of each sample was evaluated for specific genes of interest using the TaqMan Gene Expression Master Mix and TaqMan probes against the genes of interest according to the protocol provided by the manufacturer (Thermo Fisher Scientific, Waltham, MA, USA). In an independent experiment, the ΔCT values for the specific genes were calculated using β-actin as the house-keeping gene control and the 2^−ΔΔCT^ was calculated relative to the experimental control group (i.e., control siRNA transfected HRMECs treated with PBS). Each experimental group had an equal sample size from two independent experiments. In an independent experiment, each group had triplicates and was evenly distributed across different cultureware. Therefore, the 2^−ΔΔCT^ for each group was analyzed with a mixed-effects linear regression model with only independent experiments as a random effect using Stata-17 software (StataCorp LLC, College Station, TX, USA). Results are presented as mean ± standard error (SE) and a *p*-value < 0.05 was considered statistically significant.

## Figures and Tables

**Figure 1 ijms-23-07354-f001:**
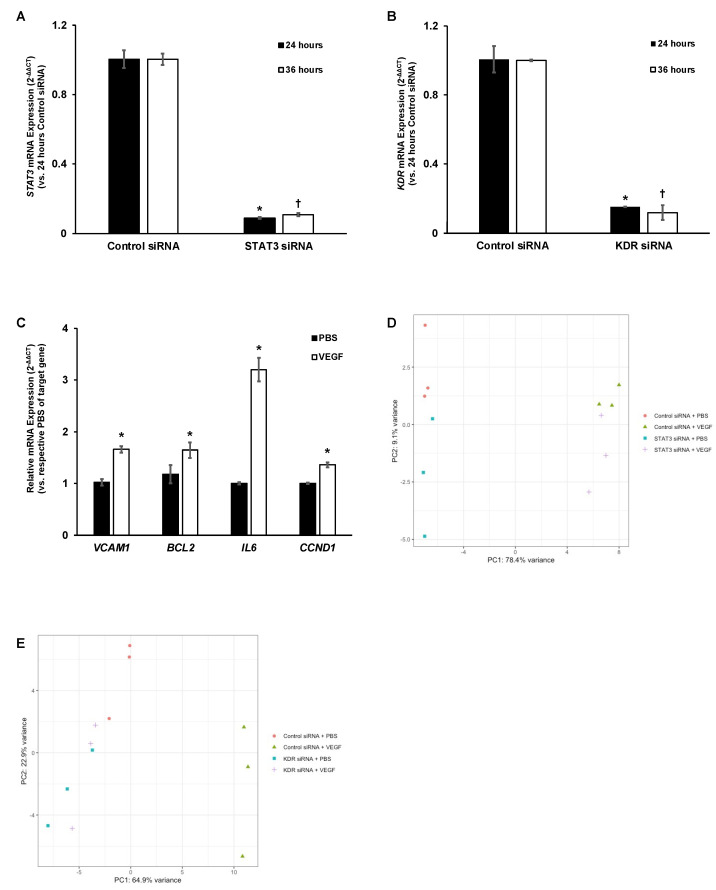
Verification of treatment conditions used in RNA-seq experiments. Sufficient knockdown of (**A**) *STAT3* or (**B**) *KDR* mRNA was observed 24–36 h after transfection in cultured HRMECs (normalized mean ± SEM, * *p* < 0.05 vs. 24 h Control siRNA-transfected HRMECs, † *p* < 0.05 vs. 36 h control siRNA-transfected HRMECs, *n* = six per group from two independent experiments); VEGF-mediated gene expression changes of (**C**) *VCAM1*, *BCL2*, *IL6*, and *CCND1* were observed after four hours of VEGF treatment compared to PBS, vehicle control, treatment in HRMECs (normalized mean ± SEM, * *p* < 0.05 vs. respective PBS, *n* = six per group from two independent experiments); PCA identified the grouping of transcriptomes from HRMECs treated with similar conditions in the (**D**) STAT3 RNA-seq dataset (*n* = three biologic replicates per group from the same independent experiment) and (**E**) KDR RNA-seq dataset (*n* = three biologic replicates per group from the same independent experiment).

**Figure 2 ijms-23-07354-f002:**
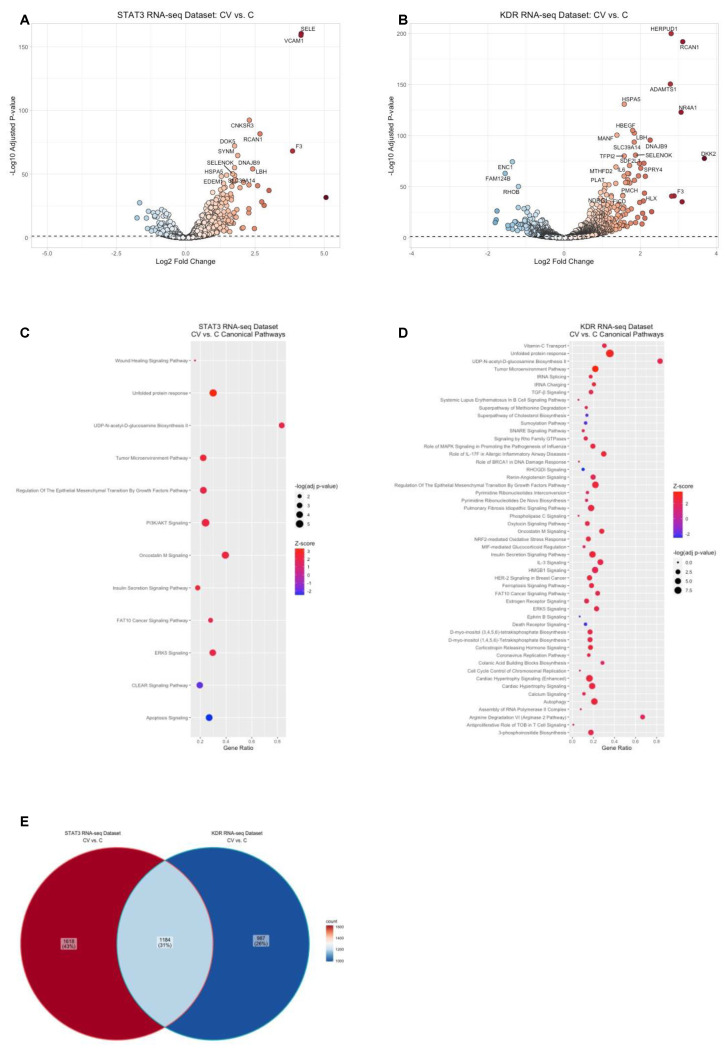
VEGF-mediated gene regulation and pathway enrichment in HRMECs. Visualization of the VEGF-mediated genes (VEGF-treated control siRNA-transfected HRMECs (CV) group compared to the PBS-treated control siRNA-transfected HRMECs (C) group) by volcano plot from (**A**) the STAT3 RNA-seq dataset or (**B**) the KDR RNA-seq dataset (black dashed line represents significance threshold value of −log(adjusted *p*-value) = 1.30, color gradient visually depicts the fold of gene upregulation (red) and downregulation (blue) in the CV group compared to the C group); bubble plots depicting the significantly enriched and significantly activated or inactivated pathways associated with the VEGF-mediated genes from (**C**) the STAT3 RNA-seq data or (**D**) the KDR RNA-seq dataset (size of bubble directly correlates with the significance of pathway enrichment, color gradient visually depicts the predicted strength of activation (red) or inhibition (blue) of enriched pathways by IPA, gene ratio (*x*-axis label) was calculated by taking the number of genes from the uploaded dataset that was associated with the Canonical Pathways (*y*-axis labels), and dividing it by the total number of genes from the IPA knowledge database associated with the respective Canonical Pathway); (**E**) visualization of the common or unique VEGF-mediated genes between the two RNA-seq datasets by a Venn diagram (color gradient visually depicts relatively higher number of gene counts (red) or relatively lower number of gene counts (blue).

**Figure 3 ijms-23-07354-f003:**
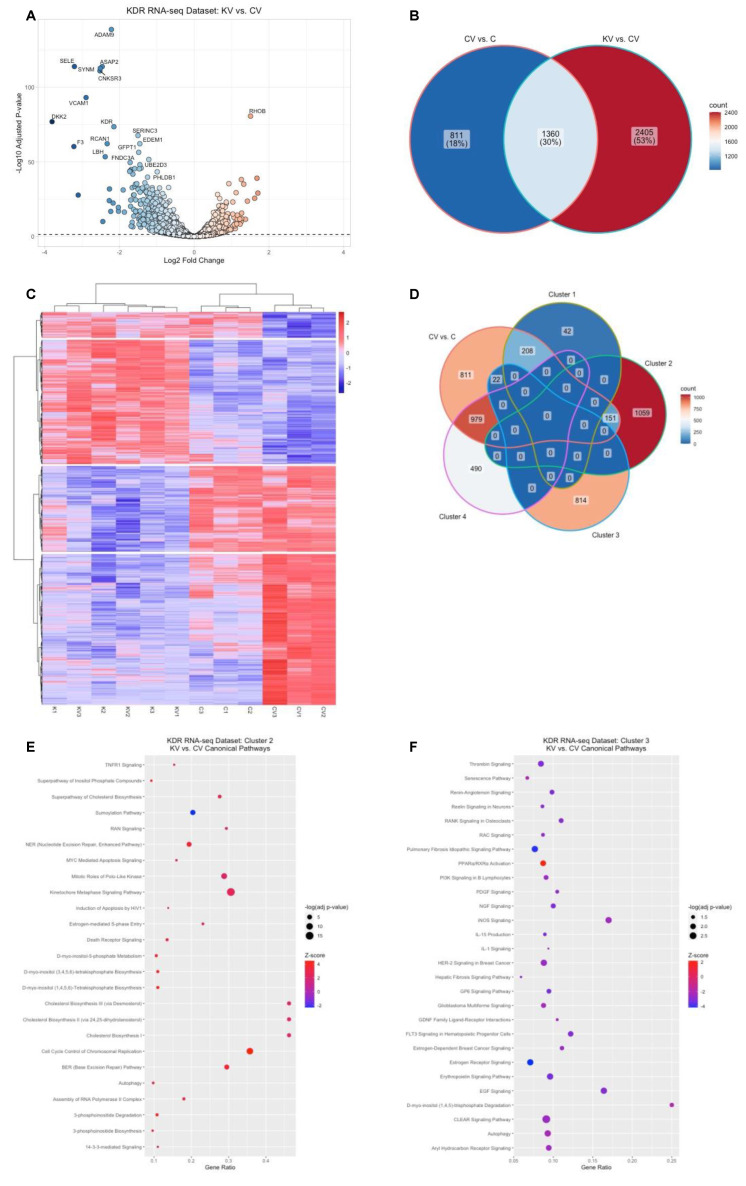
VEGFR2-mediated gene regulation and pathway enrichment in HRMECs. (**A**) Visualization of the VEGF-induced VEGFR2-mediated genes (VEGF-treated KDR siRNA-transfected HRMECs (KV) group compared to the VEGF-treated control siRNA-transfected HRMECs (CV) group) by volcano plot from the KDR RNA-seq dataset (black dashed line represents significance threshold value of −log(adjusted *p*-value) = 1.30; the color gradient visually depicts the fold of gene upregulation (red) and downregulation (blue) in the KV group compared to the CV group); (**B**) visualization of the common genes between the VEGF-mediated genes (i.e., CV vs. C) and the VEGF-induced VEGFR2-mediated genes (KV vs. CV) by a Venn diagram (color gradient visually depicts the relatively higher number of gene counts (red) or relatively lower number of gene counts (blue); (**C**) hierarchical clustering of the VEGF-induced VEGFR2-mediated genes identified four different clusters of gene expression patterns (row dendrogram, color gradient represents higher (red) or lower (blue) Z-score values of the RLog transformed read counts for each biologic sample (*n* = 12, column dendrogram) relative to each gene); (**D**) Venn diagram depicting the overlap of VEGF-mediated genes (CV vs. C) with the VEGF-induced VEGFR2 (KV vs. CV) hierarchical clusters; bubble plots depicting the significantly enriched and significantly activated or inactivated pathways associated with the VEGF-induced VEGFR2-mediated genes from (**E**) cluster 2 or (**F**) cluster 3 (size of bubble directly correlates with the significance of pathway enrichment, color gradient visually depicts the predicted strength of activation (red) or inhibition (blue) of enriched pathways by IPA, gene ratio (*x*-axis label) was calculated by taking the number of genes from the uploaded dataset that was associated with a canonical pathway (*y*-axis labels), and dividing it by the total number of genes from the IPA knowledge database associated with that respective canonical pathway.

**Figure 4 ijms-23-07354-f004:**
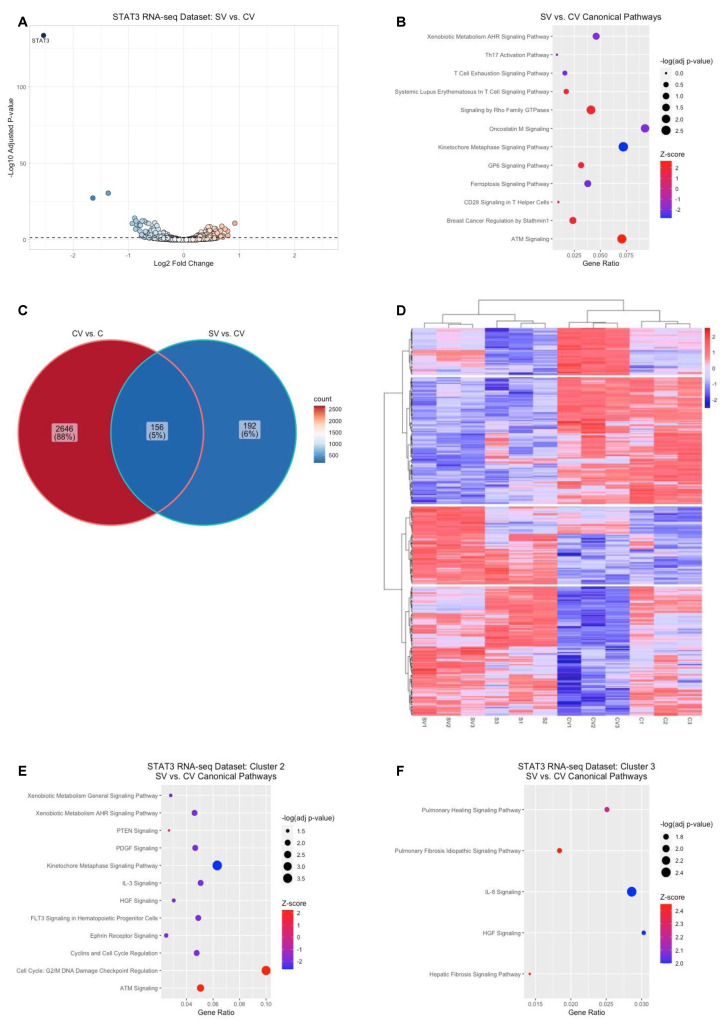
STAT3-mediated gene regulation and pathway enrichment in HRMECs. (**A**) Visualization of the VEGF-induced STAT3--mediated genes (VEGF-treated STAT3 siRNA-transfected HRMECs (SV) group compared to the VEGF-treated control siRNA-transfected HRMECs (CV) group) by volcano plot from the STAT3 RNA-seq dataset (black dashed line represents significance threshold value of −log(adjusted *p*-value) = 1.30, color gradient visually depicts the fold of gene upregulation (red) and downregulation (blue) in the SV group compared to the CV group); (**B**) bubble plot depicting the significantly enriched and significantly activated or inactivated pathways associated with the VEGF-induced STAT3-mediated genes (size of bubble directly correlates with the significance of pathway enrichment, color gradient visually depicts the predicted strength of activation (red) or inhibition (blue) of enriched pathways by IPA, gene ratio (*x*-axis label) was calculated by taking the number of genes from the uploaded dataset that was associated with a canonical pathway (*y*-axis labels); (**C**) visualization of the common genes between the VEGF-mediated genes (i.e., CV vs. C) and the VEGF-induced STAT3-mediated genes (SV vs. CV) by a Venn diagram (color gradient visually depicts the relatively higher number of gene counts (red) or relatively lower number of gene counts (blue); (**D**) hierarchical clustering of the VEGF-induced STAT3-mediated genes identified four different clusters of gene expression patterns (row dendrogram, color gradient represents higher (red) or lower (blue) Z-score values of the RLog transformed read counts for each biologic sample (*n* = 12, column dendrogram) relative to each gene); bubble plots depicting the significantly enriched and significantly activated or inactivated pathways associated with the VEGF-induced STAT3-mediated genes from (**E**) cluster 2 or (**F**) cluster 3 (size of bubble directly correlates with the significance of pathway enrichment, color gradient visually depicts the predicted strength of activation (red) or inhibition (blue) of enriched pathways by IPA, gene ratio (*x*-axis label) was calculated by taking the number of genes from the uploaded dataset that was associated with a canonical pathway (*y*-axis labels), and dividing it by the total number of genes from the IPA knowledge database associated with that respective canonical pathway.

## Data Availability

The data that support the findings of this study are available from the corresponding author, M.E.H., upon reasonable request.

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
