# Peer review of "RNA-Seq Provides Insights into VEGF-Induced Signaling in Human Retinal Microvascular Endothelial Cells: Implications in Retinopathy of Prematurity"

_ijms, 2022, doi:10.3390/ijms23137354_

Round 1
Reviewer 1 Report
The authors have addressed the vascular endothelial growth factor (VEGF) signaling in endothelial cells by knocking down its primary receptor, VEGFR2 or STAT3 and then exploring the subsequent differences caused by these knockdowns in the RNA-seqs. Then they have compared the different RNA-seq datasets by performing pathway enrichment analysis to identify signaling pathways IN VEGF-signaling influences by either VEGFR2/STAT3.
The experiments have been carefully conducted, the conclusion supported by the data and the manuscript is well written. I have no further comment.
Author Response
We thank the reviewer for taking the time to carefully review our manuscript.
Reviewer 2 Report
General comments:
Retinopathy of prematurity (ROP) is a condition that occurs in babies born prematurely in which abnormal blood vessels develop in the back of the eye. ROP involves delayed intraretinal vascularization followed by intravitreal neovascularization (IVNV). The main objective of this manuscript is to test the hypothesis that there are distinct pathways for VEGF-triggered VEGFR2 signaling and VEGF-triggered STAT3 signaling in human retinal microvascular endothelial cells (HRMECs). Key findings include: 1) In enriched pathways, inactivation of oncostatin M signaling is predicted by KDR or STAT3 knockdown in the presence of VEGF; 2) KDR knockdown predicts activation of kinetochore metaphase signaling, whereas inactivation was predicted by STAT3 knockdown in the presence of VEGF; and 3) KDR knockdown predicted inactivation of Rho family GTPase signaling, but activation was predicted by STAT3 knockdown in the presence of VEGF. While some of the data is interesting, there are still some major concerns that need to be addressed.
Comments to the authors:
Major comments:
- The main hypothesis of this manuscript is based on the results of previous study in the rat oxygen-induced retinopathy (OIR) model. However, the main results of this manuscript were generated by human retinal microvascular endothelial cells. How do the authors apply the results from the rat model to human studies? In other words, can some of the key findings of this manuscript be replicated in rat retinal microvascular endothelial cells?
- In this manuscript, they used human retinal microvascular endothelial cells (HRMECs) in paragraphs 3-5. As we all know, primary HRMEC needs to be used within 3 paragraphs. HRMECs may lose their phenotype in passages 4 to 5. The authors need to address this issue.
- In this manuscript, the authors used VEGF at a concentration of 25 ng/mL. What is the rationale for using this concentration? This reviewer wondered if they found a dose effect of VEGF in Figure 1.
- In lines 566-567, the authors found that in the presence of VEGF, STAT3 knockdown resulted in differential expression of 348 genes; however, only these genes were also differentially expressed by VEGF in the same dataset. As interesting as the data are, the reviewer wonders what the translational implications of this finding are? In other words, based on this finding, can we find new targets for retinopathy of prematurity? The authors need to address these issues in the manuscript.
